# Cardio-Respiratory Fitness and Fatigue in Post-COVID-19 Syndrome—A Three-Year Update

**DOI:** 10.3390/biomedicines13051097

**Published:** 2025-04-30

**Authors:** Radostina Cherneva, Zheyna Cherneva, Vania Youroukova, Tanya Kadiyska, Dinko Valev, Ebru Myuyun Hayrula-Manaf, Vanyo Mitev

**Affiliations:** 1Respiratory Intensive Care Unit, University Hospital “St Ivan Rilski”, Medical University of Sofia, 1431 Sofia, Bulgaria; 2Clinic of Cardiology, Hospital of Ministry of Interior, 1431 Sofia, Bulgaria; jenicherneva@yahoo.com; 3Clinic of Respiratory Diseases, University Hospital “St Ivan Rilski”, Medical University of Sofia, 1431 Sofia, Bulgaria; vania_youroukova@hotmail.com; 4Department of Physiology and Pathophysiology, Medical University of Sofia, 1431 Sofia, Bulgaria; kadiyska_t@yahoo.com; 5Clinic of Internal Diseases, First University Hospital “St John—The Baptist”, Medical University of Sofia, 1431 Sofia, Bulgaria; dinko.g.valev@abv.bg (D.V.); ebru.hayrula5@gmail.com (E.M.H.-M.); 6Chemistry and Biochemistry, Medical University of Sofia, 1431 Sofia, Bulgaria; vmitev@mu-sofia.bg

**Keywords:** long-term post-COVID-19, chronic fatigue, cardio-respiratory fitness, cardio-pulmonary exercise testing

## Abstract

**Background:** Post-COVID-19 syndrome (PCS) is defined as the persistence of symptoms 3 months after acute SARS-CoV-2 infection. The long-term prevalence and clinical progression of PCS has not been established. Our aim was to investigate the symptoms in PCS patients, explore the degree of physical activity, according to the fatigue severity score, and analyze its association with basic cardio-pulmonary exercise testing (CPET) parameters. **Methods:** A total of 192 subjects with history of SARSCoV-2 infection were included. They filled in the Chronic Fatigue Syndrome Questionnaire (CFSQ) and were divided into symptomatic and asymptomatic groups. Forty-seven had persistent post-COVID complaints—reduced physical capacity, fatigue, dyspnea, sleep disturbances, muscle pain. CPET was performed and the pathophysiological parameters in the different fatigue severity groups were compared. **Results:** Subjects with persistent long-term PCS were divided into two groups—mild (20) and moderate–severe (27), depending on the CFSQ score; forty-eight PCS subjects without complaints served as a control group. The average period between the acute illness and the study was 1028 ± 214 days. Subjects with moderate–severe PCS had more symptoms during CPET (73.6% vs. 24.8% vs. 17.4%), as compared to mild/asymptomatic. The rate of perceived effort was subjective and did not correspond to the workload, heart, or breathing rate in the symptomatic group. These subjects were unable to reach the anaerobic threshold, compared to mild/asymptomatic subjects (51.8% vs. 25%, vs. 12.5%). Patients with moderate–severe PCS showed lower peak VO_2_ (24.13 ± 6.1 mL/min/kg vs. 26.73 ± 5.9 mL/min/kg, vs. 27.01 ± 6.3 mL/min/kg), as compared to mild/asymptomatic subjects. **Conclusions:** Long-term PCS is still present in up to 24% of the general population, more than thirty months after the acute episode. It is characterized by increased perception of symptom burden and diminished aerobic metabolism. A third of the long-term PCS exhibit lower cardio-respiratory fitness, independently from the severity of the symptoms.

## 1. Introduction

A novel coronavirus SARS-CoV-2 (COVID-19) appeared in late 2019 and was followed by an outbreak of pandemic coronavirus disease. The clinical picture of the disease is quite variable, depending on the affected organ/system—ranging from acute respiratory illness and peri-myocarditis to encephalitis, visceral thrombosis, or rare bacterial and mycotic pulmonary infections [1,2,3]. A lot of research has focused on the long-term sequelae of COVID-19; several reports demonstrate the presence of symptoms in subjects recovering from COVID-19, even in those with milder disease [4]. Similarly, other chronic post-viral fatigue syndromes, characterized by chronic fatigue, myalgia, depression, and sleep disturbances, have previously been reported [5,6].

The World Health Organization (WHO) defines the post-acute sequelae of the coronavirus 2 (SARS-CoV-2) infection, also known as post-COVID-19 syndrome (PCS), as the presence of symptoms for more than three months after the acute SARS-CoV-2 infection [7]. It is estimated to affect at least 65 million individuals worldwide [8]. More than 80% of hospitalized and 40% of the non-hospitalized patients may experience symptoms for longer than 12 months [9]. Among them, most common are those responsible for the limited physical activity and exercise capacity, including fatigue, dyspnea, decreased muscle strength, muscle pain, neuro-cognitive problems, and sleep disturbances [10,11]. Several predictive risk factors for the development of post-COVID physical exhaustion have been identified: female sex, identified pre-existing conditions (cardiovascular, respiratory, neurological, oncological or autoimmune), the length of hospital stay, the severity of illness (although milder courses can also trigger PCS) [12,13]. Despite this, the prevalence of PCS in the long term has not been studied, nor have its pathophysiological mechanisms and detrimental consequences on physical function and fatigue been elucidated. While generalized treatments and recommendations may be applied, specific PCS therapy does not yet exist [7]. Therefore, unraveling the underlying pathophysiological mechanisms is essential for future targeted treatment.

Cardio-pulmonary exercise testing (CPET) is considered the “gold standard” for identifying cardiovascular, ventilatory, and musculo-skeletal limitations during exercise, and is considered as one of the most effective non-invasive methods for a complete analysis of diminished physical activity in the general population, as well as in subjects with PCS [14].

We aimed to demonstrate the prevalence of long-term PCS, three years after acute illness, and also described the basic pathophysiological abnormalities in subjects with mild and moderate–severe post-COVID fatigue. A complete assessment of the main CPET parameters for physical capacity in long-term PCS subjects was performed.

## 2. Material and Methods

We performed a cross-sectional study among outpatient workers of a private Electric Holding company, that were clinically stable, without any acute or chronic illnesses or complications. During the pandemic, workers were routinely tested for infection before going to work; the presence of SARS-CoV-2 infection was reported and validated in their medical history. The recruitment period was between January 2024–March 2024. The study followed the recommendations for medical research in the Declaration of Helsinki, Good Clinical Practice standards, and was approved by the local Ethical Committee at the Medical University, Sofia (protocol 2417/28 May 2023). All the subjects were acquainted with the aim of the study, its scientific value and the potential presentation of data at different forums. They all signed informed consent before participation.

The inclusion criteria were (1) patients with a SARS-CoV-2 infection episode that had occurred at least 12 months ago; (2) subjects willing to undergo CPET. The following exclusion criteria were considered: (1) left ventricular ejection fraction (LVEF) < 50%; (2) the presence of echocardiographic criteria of pulmonary hypertension (systolic pulmonary arterial pressure > 36 mmHg, maximum velocity of the tricuspid regurgitation jet > 2.8 m/s); (3) valvular heart disease; (4) documented cardiomyopathy; (5) severe, uncontrolled hypertension (systolic blood pressure > 180 mmHg and diastolic blood pressure > 90 mmHg); (6) atrial fibrillation or malignant ventricular arrhythmia; (7) recent chest or abdominal surgery; (8) recent exacerbation (during the last three months) of asthma or chronic obstructive pulmonary disease; (9) fatigue must not be the result of an psychiatric/neurological disease (depression, anxiety, fibromyalgia, sleep disorders, neurodegenerative disorders); infectious diseases (herpes simplex virus, enterovirus, Lyme disease, Q fever), endocrine disease (hypothyroidism, diabetes mellitus, severe obesity); immunologic disorders (lupus, multiple sclerosis, temporo-mandibular joint disorders) (Box 1).

Patients with history of COVID-19 at least, 12 months ago were asked to complete the Chronic Fatigue Syndrome Questionnaire (CFSQ). A total of 47 of 192 subjects had long-term complaints and were invited for cardio-pulmonary exercise testing. Based on the CFSQ score, subjects were additionally divided into two groups—subjects with mild complaints (CFSQ score: 10–25 points) and those with moderate–severe complaints (CFSQ score: 25–50) [15]. A total of 48 sex- and age-matched post-COVID subjects with no complaints served as a control group.

Box 1Inclusion and exclusion criteria.Inclusion criteria:(1)patients with SARS-CoV-2 infection episode that has occurred at least 12 months ago;(2)subjects who are willing to undergo CPET.Exclusion criteria:(1)left ventricular ejection fraction (LVEF) < 50%;(2)presence of echocardiographic criteria of pulmonary hypertension (systolic pulmonary arterial pressure > 36 mmHg, maximum velocity of the tricuspid regurgitation jet > 2.8 m/s);(3)valvular heart disease;(4)documented cardiomyopathy;(5)severe uncontrolled hypertension (systolic blood pressure > 180 mmHg and diastolic blood pressure > 90 mmHg);(6)atrial fibrillation or malignant ventricular arrhythmia;(7)recent chest or abdominal surgery; (8)recent exacerbation (during the last three months) of asthma or chronic obstructive pulmonary disease; (9)fatigue must not be the result of an psychiatric/neurological disease (depression, anxiety, fibromyalgia, sleep disorders, neurodegenerative disorders); infectious diseases (herpes simplex virus, enterovirus, Lyme disease, Q fever), endocrine disease (hypothyroidism, diabetes mellitus, severe obesity); immunologic disorders (lupus, multiple sclerosis, temporo-mandibular joint disorders).

### 2.1. Procedures

#### 2.1.1. Pulmonary Function Testing

The symptomatic subjects, as well as the control group, were instructed to refrain from smoking, caffeine, alcohol ingestion, and intensive physical activity on the day of testing, and were to eat a light breakfast only. As per protocol, spirometry was carried out first, in order to estimate the minute voluntary ventilation and evaluate the breathing reserve. Spirometry and CPET were performed on the same device—Vyntus, cardio-pulmonary exercise testing (Carefusion, Hoechberg, Germany) in accordance with ERS guidelines [16].

#### 2.1.2. Stress Test Protocol—Cardio-Pulmonary Exercise Testing (CPET)

A symptom-limited incremental exercise stress test was performed, following the guidelines [17]. It was performed on Vyntus, cardio-pulmonary exercise testing (Carefusion, Hoechberg, Germany). Subjects were instructed to keep a speed of 60–65 rotations per minute. A continuous ramp protocol was applied: rest phase—0 W/; warm-up phase—20 W/3 min; test phase—20 W/2 min load increments; recovery phase—0 W/3 min. Effort was considered as maximal if two of the following criteria emerged: predicted maximal HR is achieved; predicted maximal work is achieved; ‘VE/‘VO_2_ > 45, RER > 1.10, as recommended by the ATS/ACCP [18].

Expiratory gas evaluation was performed by using a breath-by-breath analysis. The highest 30 s average value, obtained during the last stage of the exercise test, was assumed as a peak value for oxygen consumption and carbon dioxide production. The peak respiratory exchange ratio was the highest 30 s averaged value between ‘VO_2_ and ‘VCO_2_ during the last stage of the test. Ten-second averaged ‘VE and VCO_2_ data, from the initiation of exercise to peak, were used to calculate the ‘VE/‘VCO_2_ slope, via least-squares linear regression. This was used to present the ventilatory response at peak exercise.

Measurements included heart rate (HR), blood pressure, arterial blood oxygen saturation (SaO_2_), oxygen consumption (VO_2_), carbon dioxide production (VCO_2_), and minute ventilation (VE). ‘VO_2_ (mL/kg/min), ‘VCO_2_ (L/min), ‘VE (L/min), and PetCO_2_ (mmHg) were obtained subsequently—at rest and throughout the whole exercise test. Exercise capacity was defined as normal when the peak VO_2_ was greater than or equal to _80_% of that predicted. Limitation in functional capacity was classified as mild (VO_2_ peak if between 65 and 80%), moderate (VO_2_ peak between 50 and 65%), and severe (VO_2_ peak less than 50%), respectively. The reasons for the diminished exercise capacity were described as cardiovascular, respiratory, peripheral, or mixed. The participants’ rate of perceived exertion (RPE) was evaluated using the modified BORG Scale, serving as a subjective indicator of intensity. It ranges from 6 to 20, with 6 meaning “no exertion at all” and 20 meaning “maximal exertion”. The number that best describes the perceived level of exertion during physical activity is chosen by the individual.

Peak VO_2_ is expressed as mL/kg body weight/min and is also presented as a percentage of predicted. VO_2_ at the anaerobic threshold (AT) was identified by the ventilation-slope method and is described as the cross-point between the ventilatory equivalent for oxygen (VE/VO_2_), without a concomitant increase in the ventilatory equivalent for carbon dioxide (VE/VCO_2_).

Breathing reserve was calculated as: (MVV − peak V’E)/MVV ∗ 100; MVV is the maximal voluntary ventilation estimated as FEV1 multiplied by 35.

### 2.2. Statistical Analysis

STATA 13.0 software packages were used for statistical analysis. A value of *p* < 0.05 was considered statistically significant. Continuous variables were expressed as median and interquartile range when data were not normally distributed, and with mean ± SD if normal distribution was observed. Categorical variables were presented as number (*n*) and percentages (%). The Kolmogorov–Smirnov test was used to explore the normality of distribution. Depending on its presence or absence, the comparisons of continuous variables between the two groups (patients with/without post-PASC) was performed by Student’s *t* test or Mann–Whitney U test, respectively. To compare continuous variables between mild long-term post-COVID, moderate–severe long-term post-COVID and the control group, ANOVA or Kruskal–Wallis H-tests were applied, considering the distribution of the parameters of interest. The association between CPET parameters and long-term post-COVID-19 syndrome was determined by univariate analysis. The same method was applied to determine predictors for long-term PCS.

A power analysis during the study was performed, on the basis of the difference in the parameters that were of investigational interest, applying a two-tailed test with a level of significance—0.05. We obtained different values of power within a range of 65–80%. Thus, we established the power of the study as sufficient.

## 3. Results

### 3.1. Participants’ Characteristics

The anthropometric and clinical characteristics of the patients are shown in Table 1. A total of 192 subjects (mean age 44.38 ± 7.6 years; 62% men) participated in the study. The average period between the initial diagnosis of COVID-19 and the time of cardio-pulmonary exercise testing was 1028 ± 214 days. Forty-seven of one hundred and ninety-two subjects had long-term post-COVID-19 syndrome based on CFSQ score. They were additionally divided according to the degree of CFSQ into mild (*n* = 20) and moderate–severe (*n* = 27). Forty-eight post-COVID subjects without complaints served as a control group.

Comorbidities already present prior to COVID-19 infection in symptomatic patients include cardiovascular disease—12 (26%); depression—2 (4%); diabetes—6 (13%); arterial hypertension 14 (29%) COPD—0; asthma—2 (4%); restrictive lung disease—0; renal insufficiency—0; The smoker–nonsmoker ratio was 21:26. In the asymptomatic group, there was no significant difference regarding comorbidity: cardiovascular disease—14 (29%); depression—1 (2%); diabetes—5 (10%); arterial hypertension—15 (31%) COPD—0; asthma—4 (8%); restrictive lung disease—0; renal insufficiency—0. The smoker–nonsmoker ratio was 23:25.

Patients from the symptomatic group had concomitant medication: ACE inhibitors—14 (30%); ß-blockers—8 (17%); anti-obstructive medications—2 (4%); antidepressants 2 (4%), antidiabetics—4 (8%); anticoagulants—8 (17%); aspirin—8 (17%); a total of 21 patients were not taking any medications.

Out of 49 asymptomatic patients, ACEI were found in 14 (29%); ß-blockers—10 (20%); anti-obstructive medications—4 (8%); antidepressants—1 (2%); antidiabetics—4 (8%); anticoagulants—4 (8%); aspirin—8 (16%); a total of 28 patients were not taking any medications. Data regarding vaccination status were not collected.

### 3.2. Cardio-Pulmonary Parameters

The cardio-pulmonary parameters of the study group are presented in Table 2. Patients with moderate–severe long-term PCS showed lower peak VO_2_ (24.13 ± 6.1 mL/min/kg vs. 26.73 ± 5.9 mL/min/kg vs. 27.01 ± 6.3 mL/min/kg), as compared to mild and asymptomatic subjects, but this was not statistically significant. Regarding oxygen consumption, given as the predicted value, 10 (37%) of the moderate–severe subjects had mildly diminished functional capacity with average peak VO_2_ (77.9% ± 8.8%). None had moderate or severe capacity limitation. Sixty-three percent had preserved physical activity—VO_2_ (85.9% ± 8.7%), despite being highly symptomatic. Only 7 (35%) of the mildly symptomatic subjects had limited physical capacity—VO_2_ (81.7 ± 6.9%); the rest had preserved—VO_2_ (87.7 ± 6.2%). The physical capacity of the asymptomatic patients was limited in 14 (29%)—VO_2_ (83.7 ± 2.3), while 34 (71%) showed normal results—VO_2_ (90.7 ± 4.3%).

The moderate–severe long-term PCS could not achieve the AT in 51.8% of the tests. The mildly symptomatic subjects could not succeed in doing so in 25% of the tests, while the asymptomatic PCS subjects failed in only 12.5% of the cases. Moderate–severe long-term PCS patients were more symptomatic compared to the mildly symptomatic group (73.6% vs. 24.8%), as well as to the asymptomatic group (73.6% vs. 17.4). Dyspnea was predominant in 77.8%; leg fatigue—22.2%; chest pain—0%. See Table 3.

We investigated whether certain CPET parameters and PCS (Table 4) can serve as reliable biomarkers for long-term PCS (Table 5).

## 4. Discussion

We explored the characteristic features of long-term PCS patients according to their fatigue severity score. The main findings of the study are as follows: (1) 24% of the patients had long-term PCS; (2) moderate–severe PCS was more symptomatic and had a lower threshold for perception of dyspnea or leg fatigue in comparison to the mildly symptomatic group; (3) only 37% of the moderate–severe and 35% of the mildly symptomatic long-term PCS patients exhibited objectively reduced physical capacity and oxygen consumption; (4) a statistically significant part of the moderate–severe patients did not reach the anaerobic threshold.

Post-COVID-19 syndrome includes persistent symptoms that may be related to residual inflammation, organ damage or non-specific effects from treatment, or the impact of pre-existing health conditions [19]. In this study, we explored the long-term PCS-related symptoms, defined as low cardio-respiratory fitness—dyspnea or fatigue, persisting for at least 30 months after symptom onset. According to our results, 24% of the patients exhibited long-term PCS; fatigue and limited cardio-respiratory fitness were the most common subjective complaints.

Several studies have reported the prevalence of persistent symptoms, usually up to 12 months after disease onset. It varies from 40 to 90% after hospital discharge. Most common complaints include fatigue—53%; breathlessness—27%; and joint pain—16% [20]. Halpin et al. reported that new illness-related fatigue was the most reported symptom—in 72% of participants who required intensive care and 60.3% of the non-intensive care subjects; breathlessness was reported in 65.6% of patients in intensive care units and 42.6% of those who were in hospital wards [21].

We investigated risk factors as potential contributors related to the long-term persistence of symptoms. Regarding our results, none of the CPET parameters can be considered predictors for long-term post-COVID. Obesity is commonly associated with a low cardio-respiratory fitness and has been implicated as a risk factor for both acute severe and post-COVID illness. In contrast to other authors, we found BMI does not significantly differ across fatigue severity groups, nor is a difference observed when compared to asymptomatic individuals [22,23]. Similarly to Tenforde et al., our cohort included patients who had mild-to-moderate acute COVID-19 and had not been hospitalized. Our findings correspond to their data, as we also were not able to find an association between the severity of the acute illness and the development of long-term sequalae [24].

As in other studies, the moderate–severe long-term PCS population demonstrates reduced VO_2 max_, when compared to the asymptomatic or mildly symptomatic patients. If a careful analysis is performed, the percentage of subjects with actually limited VO_2_ < 80% physical capacity is almost identical in the mild (35%) and moderate–severe (37%) group. Thus, the reduction in oxygen consumption may affect an equivalent number of long-term PCS, but with a different degree of symptom severity.

Peak oxygen consumption (VO_2_) represents the combined capacity of the respiratory, cardiovascular, and muscle systems to acquire, transport, and utilize O_2_ [25]. Thus, the decrease in peak VO_2_ observed in patients with PCS, compared to asymptomatic/mildly symptomatic subjects, could be explained by multiple factors. In contrast to the PCS that follows the acute illness, the long-term PCS may not be attributed to reduced fitness before disease onset, reduction due to COVID-19-induced physical impairments, the side effects of drug treatment, or sedentary lifestyle because of quarantine [26,27].

Clavario et al. reported that 50% of non-severe COVID-19 subjects have physical capacity limitation due to muscular impairment [28]. Barbegelata et al. did not find a specific reason (cardio-respiratory, peripheral, ventilatory or mixed) distinguishing symptomatic/asymptomatic post-COVID regarding their limited physical activity [29]. Beyer et al. reported that the leading cause for exercise test discontinuation was peripheral exhaustion [23]. Seeßle at al. detected that 40% of PCS patients have dyspnea [9].

In our study, dyspnea was predominant for exercise discontinuation. Leg fatigue was the second most common limiting symptom. During spiro-ergometry, we did not find dynamic hyperinflation, exercise-induced bronchospasm, exercise-induced desaturation (>3%) or an exhausted ventilatory reserve that could be assumed as an objective reason for specific pulmonary limitations. None of the patients exhibited a restrictive pattern in baseline spirometry. Our results are in contrast to those of Cortés-Telles et al., who suggest that patients with persistent dyspnea show greater impairments in resting and exertional pulmonary gas exchange, and have a greater indication of a restrictive pattern on spirometry [30]. Others speculate that COVID-19 patients with persistent dyspnea are more likely to adopt to a more rapid and shallow breathing pattern and higher levels of respiratory neural drive during CPET [31].

Similarly to Beyer et al. [23], we noticed a lower threshold of perception for exertion. Most of the moderate–severe subjects report a high level of self-perception of exertion on BORG scale, which does not correspond to the intensity of load, current heart or breathing rate, or metabolic equivalent (RER). Central factors may partly limit exercise capacity in these patients; neuro-cognitive disturbances may also lead to a stronger perception of exertion during exercise [32]. Autonomic nervous system (ANS) dysfunction has been largely studied as a contributor for chronic fatigue syndrome [33]. However, this organ system was largely neglected so far, as a potential underlying mechanism for exercise intolerance and low cardio-respiratory fitness in patients with PCS. Patients with PCS are largely characterized by parasympathetic excess and sympathetic withdrawl, thus supporting the ANS as the missing gear in the Wasserman gear as an independent contributing factor in PCS [34,35,36]. The contribution of abnormal peripheral muscle physiology may also be assumed, but the data are controversial, with reports stating either normal or impaired muscle function [37]. This demonstrates the need for further research in this area.

### Limitations

The major limitations of our study are (1) the relatively small sample number, which prevents the generalizability of the findings; (2) the study was performed in a Caucasian population; (3) the study is cross-sectional, obscuring the cause–effect association; and (4) the possibility of selection bias influencing the results cannot be ruled out. Thus, our findings should be interpreted with caution, until confirmed by large-scale research.

## 5. Conclusions

Up to 24% of the patients may exhibit long-term PCS, even 30 months after the acute episode. It is characterized by an increased perception of symptom burden and diminished aerobic metabolism. A third of the long-term PCS exhibit lower cardio-respiratory fitness, independent from the severity of the symptoms.

## Figures and Tables

**Table 1 biomedicines-13-01097-t001:** The anthropometric and clinical characteristics of the patients.

	Without Long-Term Post-COVID-19 (48)	With Long-Term Post-COVID-19 (47)	*p*-Value
**Anthropometric**			
Age, years	44.6 ± 7.9	44.7 ± 6.3	0.417 *
Sex, M:F	33:15	25:22	0.218 ^†^
Smoker–Non-smoker	23:25	21:26	0.849 ^†^
BMI, kg/m^2^	25.8 ± 7.9	26.1 ± 8.4	0.108 ^†^
**Comorbidities, *n* (%)**			
Arterial hypertension	15 (31)	14 (29)	0.817 ^†^
Ischaemic heart disease	14 (29)	12 (26)	0.109 ^†^
Diabetes	5 (10)	6 (13)	0.087 ^†^
Dyslipidemia	5 (10)	8 (17)	0.518 ^†^
COPD/Asthma	0/4 (0/8)	0/2 (0/4)	0.832 ^†^
Depression	1 (2)	2 (4)	0.719 ^†^
**Concomitant medication, *n* (%)**			
ACE inhibitors	14 (29)	14 (30)	0.087 ^†^
Beta-blockers	10 (20)	8 (17)	0.323 ^†^
Statins	5 (10)	8 (17)	0.513 ^†^
Anti-diabetic therapy	4 (8)	4 (8)	0.065 ^†^
Bronchodilators	4 (8)	2 (4)	0.701 ^†^
Anticoagulants	4 (8)	8 (17)	0.435 ^†^
Aspirin	8 (16)	8 (17)	0.432 ^†^
**Severity of acute COVID, *n* (%)**			
Mild/non-hospitalized	38 (79)	39 (83)	0.061 ^†^
Hospitalized	10 (21)	8 (17)	0.075 ^†^

* Unpaired *t* test; ^†^ Mann–Whitney U test; Abbreviations: BMI—Body Mass Index; The data regarding medical history and concomitant medication are based on the medical history and archive of the electric company.

**Table 2 biomedicines-13-01097-t002:** The cardio-respiratory parameters of the patients.

	Without Long-Term Post-COVID-19 (48)	With Long-Term Post-COVID-19 Mild (20)	With Long-Term Post-COVID-19 Moderate–Severe (27)	*p*-Value
**Respiratory parameters**				
FEV1, L	3.16 ± 0.87	3.25 ± 0.68	3.59 ± 0.97	0.076 ^†^
FEV1, (%)	79.54 ± 11.23	83.49 ± 8.80	89 ± 8.71	0.384 ^†^
FVC, L	3.80 ± 1.09	4.11 ± 0.99	4.46 ± 1.22	0.812 ^†^
FVC, (%)	78.36 ± 13.5	82.89 ± 8.33	90 ± 8.88	0.619 ^†^
FEV1/FVC, %	79.54 ± 11.23	79.08 ± 13.21	80.49 ± 10.32	0.705 ^†^
**Physical capacity**				
Peak VO_2_, mL/min/kg	27.01 ± 6.3	26.73 ± 5.9	24.13 ± 6.1	0.098 ^†^
Predicted peak VO_2_, %	91.2 ± 3.1	84.2 ± 6.4	81.4 ± 8.6	0.298 ^†^
Exercise time, minutes	9.4 ± 2.8	9.0 ± 2.6	8.4 ± 3.2	0.112 ^†^
Slope VE/VCO_2_	32.9 ± 7.2	33.4 ± 5.9	32.1 ± 8.1	0.068 ^†^
**Categorical parameters, n (%)**				
Preserved functional capacity	34 (71)	13 (65)	17 (63)	0.109 ^†^
Mildly diminished functional capacity	14 (29)	7 (35)	10 (37)	0.084 ^†^
Moderately diminished functional capacity	0	0	0	
Achieved anaerobic threshold	42 (87.5)	15 (75)	13 (48.2)	0.571 ^†^
Depleted respiratory reserve	0	0	0	
Heart rate reserve utilization	78.12 (71.87–3.52)	68.32 (59.28–72.45)	53.28 (47.09–60.48)	0.710 ^†^

^†^ Kruskal—Wallis H test; Abbreviations: FEV1—Forced Expiratory Volume in 1 s; FVC—Forced Vital Capacity. Data are presented as mean ± SD except for heart-rate reserve utilization—median and interquartile range.

**Table 3 biomedicines-13-01097-t003:** The exercise-limiting patterns and symptoms of the patients.

	Without Long-Term Post-COVID-19 (48)	With Long-Term Post-COVID-19 Mild (20)	With Long-Term Post-COVID-19 Moderate–Severe (27)	*p*-Value
**Diminished physical activity, *n* (%)**	14 (29)	7 (35)	10 (37)	0.409 ^†^
Cardiovascular pattern	30 (62.5)	14 (70)	22 (81.4)	0.804 ^†^
Respiratory pattern	0	0	0	0.612 ^†^
Peripheral pattern	35 (72.9)	16 (80)	21 (77.8)	0.347 ^†^
**Exercise limiting symptoms, *n* (%)**				
Dyspnea	41 (85.4)	15 (75)	21 (77.8)	0.612 ^†^
Dizziness	9 (18.7)	4 (20)	5 (18.5)	0.218 ^†^
Chest pain	0	0	0	
Leg fatigue	0	5 (25)	6 (22.2)	0.703 ^†^

^†^ Mann–Whitney U test.

**Table 4 biomedicines-13-01097-t004:** Univariate logistic regression analysis between CPET and long-term PCS.

Univaraiate Regression Analysis	*p*-Value	OR	95% CI
Peak Load, W	0.345	1.781	1.512–1.967
Peak VE, L/min	0.783	1.003	0.884–1.487
Peak V’O_2_, mL/kg/min	0.904	10.137	8.125–16.014
V’O_2_ at AT, mL/kg/min	0.120	4.827	2.321–6.321
Peak RER	0.349	0.912	0.671–1.318
VE/VCO_2_ slope	0.209	6.122	3.732–8.402
HR at rest, bpm	0.717	13.56	11.032–16.087
Peak HR, bpm	0.208	0.912	0.718–1.213
O_2_ pulse, mL/beat	0.231	0.329	0.098–2.097

Abbreviations: RER—respiratory exchange ratio; AT—anaerobic threshold; HR—heart rate.

**Table 5 biomedicines-13-01097-t005:** Univariate logistic regression analysis between clinical parameters and long-term PCS.

Univaraiate Regression Analysis	*p*-Value	OR	95% CI
Age	0.809	0.403	0.232–0.692
Sex	0.209	5.098	2.097–7.613
Smoking status	0.078	2.098	0.409–4.012
BMI	0.092	1.890	0.872–3.098
Severity of the initial illness	0.430	1.098	0.409–4.987
Hospitalization	0.076	1.090	0.941–2.102
ICU admission	0.092	2.097	0.067–2.503
Arterial hypertension	0.913	9.211	1.092–12.402
Ischaemic heart disease	0.729	11.43	2.617–12.098
Diabetes	0.824	10.028	8.009–13.209
Dyslipidemia	0.098	8.072	6.124–10.044

Abbreviations: BMI—Body Mass Index; ICU—Intensive Care Unit.

## Data Availability

The data are stored in the software of Vyntus, Carefusion, which is situated in the Therapeutic Clinic of UMBAL “St Ivan Rilski”, Medical University Sofia.

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
