# Peer review of "Cardio-Respiratory Fitness and Fatigue in Post-COVID-19 Syndrome—A Three-Year Update"

_biomedicines, 2025, doi:10.3390/biomedicines13051097_

Round 1

Reviewer 1 Report

Comments and Suggestions for Authors

My comments:

                In general, this is an interesting research. The authors aim to investigate the symptoms in Post-Covid-19 syndrome (PCS) patients, to explore the degree of physical activity according to the fatigue severity score and analyze if it is associated with basic cardio-pulmonary exercise testing (CPET) parameters. They found that the long-term PCS is still present in up to 24% of the general population more than thirty months after the acute episode. A third of the long-term PCS exhibit lower cardio-respiratory fitness, independently from the severity of the symptoms. However, there are some points must be improved.              

Major

            You stated in the discussion section that “the relatively small sample number, which prevents the generalizability of the findings”. Thus, the sample size calculation must be provided. For confirmation that the results can be interpreted with high power (0.8). If the sample size was not reaching a statistic power, this study should be change to a pilot study.   

Methods

            3.1 You stated that “It was a retrospective study, performed among stable outpatient workers of a private Electric Holding company.” But “They have all signed informed consent before participation.” was stated. Please recheck

                3.2 The reference citation of the cutoff point of Chronic Fatigue Syndrome Questionnaire (CFSQ must be provided.

                3.3 Statistical analysis: You stated that “The association between CPET parameters and long-term post-COVID-19 syndrome was determined by univariate and multivariate analysis.” But there was no the result of univariate or multivariate analysis that were presented in the manuscript.  

                3.4 There were 3 groups in the table 3, thus ANOVA and Kruskal Wallis test must be used for comparing the normal distribution and non-normal distribution, respectively.    

Results

            4.1 The p-value must be provided in table1, 2 and 3.

Discussion

            In line 232, you stated that “We tried to find risk factors, related to the long-term persistence of symptoms.” However, there was no the result of univariate or multivariate logistic analysis to identify the risk factors for long-term persistence of symptoms that were presented in the manuscript. 

Reference

            Please check typo error in the reference citation, some of them were superscript format.  

Author Response

Dear reviewer 1,

Thank you for the comments.

Major:

Comment: You stated in the discussion section that “the relatively small sample number, which prevents the generalizability of the findings”. Thus, the sample size calculation must be provided. For confirmation that the results can be interpreted with high power (0.8). If the sample size was not reaching a statistic power, this study should be change to a pilot study.

Response: According to the Bulgarian Ministry of Health official COVID-19 statistics the proportion of infected varied between 5.6% in 2020 to 13.2% in 2022 https://data.egov.bg/data/resourceView/0ce4e9c3-5dfc-46e2-b4ab-42d840caab92)Thus the following formula was used to estimate the needed sample: N=Zx P (100 – P) / ∆2

e Where Z is 1.96, P is the proportion infected (5% or 11%), âˆ† is the 5% absolute desired precision

Thus we estimated that the minimal sample size for 5% incidence  was 73 individuals and the maximal for 13% incidence was 174. Therefore a sample size of 180 should be sufficient to provide a 95% Confidence and a 5% margin of error and precision. 
The final number of recruited subjects was 192. So it is sufficient.

Methods

Comment: You stated that “It was a retrospective study, performed among stable outpatient workers of a private Electric Holding company.” But “They have all signed informed consent before participation.” was stated. Please recheck

Response: 3.1. Dear reviewer, you are right. The sentence is not precise, as the  study is actually cross-sectional. Part of the data and the on-sent of the analysis starts in the past. I have changed it to – “It is a cross-sectional  study, performed among outpatient workers of a private Electric Holding company, that were clinically stable - without any acute illness or deterioriation of a chronic one.”

Comment: 3.2 The reference citation of the cutoff point of Chronic Fatigue Syndrome Questionnaire (CFSQ must be provided.

Response: 3.2. The reference is provided – 14 and all the references are already chaged

Comment: 3.3 Statistical analysis: You stated that “The association between CPET parameters and long-term post-COVID-19 syndrome was determined by univariate and multivariate analysis.” But there was no the result of univariate or multivariate analysis that were presented in the manuscript.

Response: 3.3. The statistical section is definitely unclearly presented, thus it is rewritten.

“STATA 13.0 software packages were used for statistical analysis. A value of p < 0.05 was considered statistically significant.Continuous variables were expressed as median and interquartile range, when data was not normally distributed and with mean±SD if normal distribution was observed. Categorical variables were presented as number (n) and percentages (%). Kolmogorov-Smirnov test was used to explore the normality of distribution. Depending on its presence or absence the comparisons of continuous variables between the two groups (patients with/without post-PASC) was performed by Student’s T test or Mann-Whitney U test, respectively. To compare continuous variables between mild long-term post-COVID, moderate-severe long-term post-COVID and the control group, ANOVA or Kruskal-Wallis  H tests were applied, considering the distribution of the parameters of interest.  The association between CPET parameters and long-term post-COVID-19 syndrome was determined by univariate analysis. A power analysis during the study was performed , on the basis of the difference in the parameters that were of investigational interest, applying a two-tailed test with a level of significance -  0.05. Thus, we established the power of the study as sufficient.”

Comment: 3.4 There were 3 groups in the table 3, thus ANOVA and Kruskal Wallis test must be used for comparing the normal distribution and non-normal distribution, respectively.

Response: 3.4. This comment refers to 3.3 – Statistical analysis section is renewed.

Results:

Comment: 4.1 The p-value must be provided in table1, 2 and 3.

Response: 4.1.The p-value is provided in the tables

Discussion

Comment: In line 232, you stated that “We tried to find risk factors, related to the long-term persistence of symptoms.” However, there was no the result of univariate or multivariate logistic analysis to identify the risk factors for long-term persistence of symptoms that were presented in the manuscript. 

Response: We have included an additional table and comments in the results section.

Reference

Comment: Please check typo error in the reference citation, some of them were superscript format

Response: The references were formatted to the Journal's requirement

Reviewer 2 Report

Comments and Suggestions for Authors

The study by Cherneva et al describes longterm cardiorespiratory fitness in a retrospective study of individuals 3 years after SARS-CoV-2 infection, categorized according to remaining symptoms (Long Covid) or not.

The study is of interest and highlights the high prevalence of Long-COVID following an acute SARS-CoV-2 infection and further describe potentially important clinical features associated with it. Their are certain shortcomings associated with the current manuscript that should be improved prior to publication. Especially, the authors claim to have performed a range of statistical tests but the outcome of these tests are either missing or unclearly presented, which makes it difficult to interpret presented data and the subsequent discussion. Due to the latter, I have not included any comments on the discussion section.

INTRODUCTION

  • Ref 5 deals primarily with Covid-19 in this sentence highlighting the existence of other chronic post-viral fatigue syndromes (apart from Covid-19-related). Please refer to original work instead.
  • I would suggest using the abbreviation PASC which is more commonly used (instead of PCS)

MATERIAL AND METHODS

  • Line 77: "Stable" needs to be explained - stable in what way?
  • Line 80: The ethical review board is likely associated to a university or similar, correct? Please add this information.
  • Line 84: Were the SARS-CoV-2" infections clinically verified or subjectively determined?
  • Line 89: Point 3 occurs twice, please correct
  • Line 99: The term "only" could/should be omitted.
  • Lines 100-104: The scoring is unclear to me. 1) As I understand it a CFSQ score of 10-25 were considered MILD and those with CSFQ >25 were considered moderate-to-severe. Correct? Please clarify. 2) It is stated that 47 of 192 individuals had complaints and that 48 had no complaints. What about the remaining 97 individuals??
  • Line 107: "All eligeble subjects" - how many, n=192?
  • Lines 109-110: 1) When was spirometry performed, prior to CPET? 2) Is the methodology/equipment correctly described?
  • Line 134: I guess the term "severe" is missing before "less than 50%"?
  • Line 137: Please give reference and/or a brief description of the modified Borg scale.
  • Line 149: Please add that you present categorical variables as n (%), not only %. 

RESULTS

  • Line 164: Please add "n=" before the numbers 20 and 27, respectively, for clarity.
  • Line 167: This seems incorrect. 81 of 192 is not 85%. Please check the numbers given in this paragraph!
  • Table 1: 1) Please consider presenting these data categorized as in Table 2 & 3, would be helpful. 2) Replace "," with ".". 3) Four significant figures are given for Age and BMI, three is enough. 4) Please clarify whether any statistically significant differences were present for variables included in this table! 5) How comorbidities were defined could/should be presented in a footnote of the table.
  • Lines 174-175: Meaning?? A significant difference compared to what?
  • Table 2: Please see comments for Table 1 above, points 2, 3 and 4!
  • Lines 202-204: There seems to be a numerically big difference for "achieving the AT" between these groups, but no statistical test is presented. Please add!   
Comments on the Quality of English Language

Major parts of the text are good, but certain parts need to be edited for language improvements to enhance readability and understanding. 

Author Response

Dear reviewer 2,

Thank you for the comments.

INTRODUCTION

Comment: Ref 5 deals primarily with Covid-19 in this sentence highlighting the existence of other chronic post-viral fatigue syndromes (apart from Covid-19-related). Please refer to original work instead.

Response: Ref 5 has been changed - Løkke B,  Hansen S, Dalgaard S et al .  Long-term complications after infection with SARS-CoV-1, influenza and MERS-CoV.  Lessons to learn in long COVID? Infectious Diseases Now 2023; 53(8), 104779- 104791.

Comment: I would suggest using the abbreviation PASC which is more commonly used (instead of PCS)

Response: Thank you for the suggestion. I do not agree that the abbreviation PASC is more commonly used and this does not change the value of the research, neither the understanding of the abbreviation, because it is already stated in the manuscript

MATERIAL AND METHODS

Comment: Line 77: "Stable" needs to be explained - stable in what way?

Response: The term stable is really misleading – I mean patients without any acute illness or deterioration of a chronic one – clinically stable, otherwise said.

Comment: Line 80: The ethical review board is likely associated to a university or similar, correct? Please add this information.

Response: The Ethical Committee is indeed associated to Medical University Sofia.

Comment: Line 84: Were the SARS-CoV-2" infections clinically verified or subjectively determined?

Response: During the pandemia workers were routinely tested for infection before going to work the presence of SARS-CoV-2 infection was archived and validated in their medical history – this is added for clarification in the text.

Comment: Line 89: Point 3 occurs twice, please correct

Response: The numbers are corrected - 1) left ventricular ejection fraction (LVEF) < 50%; 2) presence of echocardiographic criteria of pulmonary hypertension (systolic pulmonary arterial pressure > 36 mmHg, maximum velocity of the tricuspid regurgitation jet > 2.8 m/s; 3) valvular heart disease; 4) documented cardiomyopathy; 5) severe uncontrolled hypertension (systolic blood pressure > 180 mmHg and diastolic blood pressure >90 mmHg);6) atrial fibrillation or malignant ventricular arrhythmia;7) recent chest or abdominal surgery; 8) recent exacerbation (during the last three months) of asthma or chronic obstructive pulmonary disease; 9) the fatigue must not be the result of an psychiatric/neurological disease (depression, anxiety, fibromyalgia, sleep disorders, neurodegenerative disorders); infectious diseases (herpes simplex virus, enterovirus, Lyme disease, Q fever), endocrine disease (hypothyroidism, diabetes mellitus, severe obesity); immunologic disorders (lupus, multiple sclerosis, temporo-mandibular joint disorders).

Comment: Line 99: The term "only" could/should be omitted.

Response: The “only” was omitted.

Comment: Lines 100-104: The scoring is unclear to me. 1) As I understand it a CFSQ score of 10-25 were considered MILD and those with CSFQ >25 were considered moderate-to-severe. Correct? Please clarify. 2) It is stated that 47 of 192 individuals had complaints and that 48 had no complaints. What about the remaining 97 individuals??

Response: Dear reviewer, you are right, regarding the scoring. The clarification for the division of patients is added. We performed the comparison among two comparable by size groups, matching them by age and sex. So the selection of these 48 people is based on age and sex ‘’ Based on the score of CFSQ,  subjects  were additionally divided into two groups – subjects with mild complaints (CFSQ score:10-25 points) and those with moderate-severe complaints (CFSQ score:25-50). 48 age-matched post-COVID subjects with no complaints served as a control group.’’

Comment: Line 107: "All eligible subjects" - how many, n=192?

Response: Dear reviewer, indeed these are not the whole sample of 192 subjects, but  the symptomatic and asymptomatic subjects, adjusted by age and sex.  The symptomatic subjects, as well as, the age and sex-matched control group of subjects, were instructed to refrain from smoking, caffeine, alcohol ingestion and intensive physical activity on the day of investigation and ate a light breakfast only. As per protocol spirometry was done first, in order to help the estimation of the minute voluntary ventilation and the evaluation of the breathing reserve. Spirometry and CPET were performed on the same device - Vyntus, Cardio-pulmonary exercise testing (Carefusion, Germany) in accordance with ERS guidelines [15].

Comment: Lines 109-110: 1) When was spirometry performed, prior to CPET? 2) Is the methodology/equipment correctly described?

Response: Regarding the performance of the spirometry and CPET - Vyntus, Carefusion, CPET combines spirometry and cardio-respiratory testing simultaneously. The reason for this is that, it is recommended to perform spirometry before CPET in order to  calculate the maximum voluntary ventilation; the inspiratory capacity, end tidal volume, dynamic hyperinflation and breathing reserve. I agree , the procedures and the consequences of their execution is neither mentioned , nor explained in the text. The text is changed to: “The symptomatic subjects, as well as, the control group of subjects, were instructed to refrain from smoking, caffeine, alcohol ingestion and intensive physical activity on the day of investigation and ate a light breakfast only. As per protocol spirometry was done first, in order to help the estimation of the minute voluntary ventilation and the evaluation of the breathing reserve. Spirometry and CPET were performed on the same device - Vyntus, Cardio-pulmonary exercise testing (Carefusion, Germany) in accordance with ERS guidelines [15].”

Comment: Line 134: I guess the term "severe" is missing before "less than 50%"?

Response: Indeed the term severe is missing. The text stays “Limitation in functional capacity was classified as: mild (VO2 peak was between 65%-80%); moderate (VO2 peak between 50 and 65%) and severe (VO2 peak less than 50%), respectively.

Comment: Line 137: Please give reference and/or a brief description of the modified Borg scale.

Response: Dear reviewer, I have given a description for the Borg scale – “The participants’ rate of perceived exertion (RPE) was evaluated using the modified Borg Scale, serving as a subjective indicator of intensity. It ranges from 6 to 20, where 6 means "no exertion at all" and 20 means "maximal exertion." A number is chosen by an individual that best describes the perceived level of exertion during physical activity.

Comment: Line 149: Please add that you present categorical variables as n (%), not only %. 

Response: Dear reviewer, you are right. – “Categorical variables were presented as number (n) and percentages (%). 

RESULTS

Comment: Line 164: Please add "n=" before the numbers 20 and 27, respectively, for clarity.

Response: Dear reviewer, I have added the number (n) for clarification – In stays in the text – “They were additionally divided according to the degree of CFSQ into: mild (n=20), moderate-severe (n=27). Forty-eight post-COVID subjects without complaints served as a control group.’’

Comment: Line 167: This seems incorrect. 81 of 192 is not 85%. Please check the numbers given in this paragraph!

Response: Dear reviewer, I have calculated the numbers regarding the two groups in the table below. As for the 192 participants it is ‘’ Regarding the initial burden of COVID-19 symptoms mild disease that did not demand hospitalization was predominant, affecting 163 (85%) of the participants. In those that have been hospitalized - 28 (97) % received oxygen therapy and only 1 (3) % had non-invasive ventilation; none of them was intubated.’’

Comment: Table 1: 1) Please consider presenting these data categorized as in Table 2 & 3, would be helpful. 2) Replace "," with ".". 3) Four significant figures are given for Age and BMI, three is enough. 4) Please clarify whether any statistically significant differences were present for variables included in this table! 5) How comorbidities were defined could/should be presented in a footnote of the table.

Comment: Lines 174-175: Meaning?? A significant difference compared to what?

Comment: Table 2: Please see comments for Table 1 above, points 2, 3 and 4!

Comment: Lines 202-204: There seems to be a numerically big difference for "achieving the AT" between these groups, but no statistical test is presented. Please add!   

Response: All the other comments are referred to the presentation of the data in the tables, as well as, clarification of the statistical test which is presented in the tables.

Reviewer 3 Report

Comments and Suggestions for Authors

March 26, 2025

Ms. Ref. No.: biomedicines-3560601

Journal: Biomedicines.

Title: Cardio-respiratory fitness and fatigue in Post-Covid-19 syndrome – a three-year update

Comments:

Thank you for your efforts in composing on such a pertinent subject. I have taken the liberty of providing you with a few observations that I believe will serve to enhance the quality of your work. Please find my feedback outlined in the following paragraphs

  • The sample size of this study was 192 subjects with a history of SARSCoV-2 infection. How was determined this sample size?
  • Additionally, is this size sufficient? If yes, why?
  • Participants were split into symptomatic and asymptomatic groups. Are the numbers of these groups the same? Is that ok?
  • The typical duration between the onset of the acute illness and the study was 1028±214 days. What was the main reason for calculating this time range?
  • In the conclusion it was mentioned “Long-term PCS is still present in up to twenty-four percent of the general population more than thirty months after the acute episode”. If this study was continued after thirty months, could be change the findings be changed? Why?
  • What were the main inclusion and exclusion criteria of this study? Please add an inclusion and exclusion criteria diagram.
  • Including inclusion and exclusion criteria in a diagram seems to be better. Please add a diagram about inclusion and exclusion criteria.
  • Due to Tables 1 and 2, is the role of gender important in this study?
  • According to the Tables mentioned in this article, please check the style of the Tables of this article with the author's guideline of
  • In order to improve the clarity of the introduction, it is recommended that you include some of the following sources as references:
  • https://doi.org/10.3390/biomedicines13020516
  • https://doi.org/10.3390/biomedicines13020360
  • https://doi.org/10.3390/biomedicines12122831
  • https://doi.org/10.3390/diagnostics15060781
  • https://doi.org/10.3390/pathogens14030262s

Author Response

Dear reviewer 3,

Comment: The sample size of this study was 192 subjects with a history of SARSCoV-2 infection. How was determined this sample size?Additionally, is this size sufficient? If yes, why?

Response: According to the Bulgarian Ministry of Health official COVID-19 statistics the proportion of infected varied between 5.6% in 2020 to 13.2% in 2022 (източник https://data.egov.bg/data/resourceView/0ce4e9c3-5dfc-46e2-b4ab-42d840caab92).

Thus the following formula was used to estimate the needed sample: N=Zx P (100 – P) / ∆2

Where Z is 1.96, P is the proportion infected (5% or 11%), ∆ is the 5% absolute desired precision

Thus we estimated that the minimal sample size for 5% incidence  was 73 individuals and the maximal for 13% incidence was 174. Therefore a sample size of 180 should be sufficient to provide a 95% Confidence and a 5% margin of error and precision. The final number of recruited subjects was 192. So it is sufficient.

Comment: Participants were split into symptomatic and asymptomatic groups. Are the numbers of these groups the same? Is that ok?

Response: Dear reviewer, we tried to make an age- and sex-matched control group that is comparable by size,as well as, these major biological parameters.

Comment: The typical duration between the onset of the acute illness and the study was 1028±214 days. What was the main reason for calculating this time range?

Response: Dear reviewer, the time period I have mentioned is just a fact that I think gives a more precise overview of the clinical significance of the study, regarding the clinical presentation of long-term PSC.

Comment: In the conclusion it was mentioned “Long-term PCS is still present in up to twenty-four percent of the general population more than thirty months after the acute episode”. If this study was continued after thirty months, could be change the findings be changed? Why?

Response: Dear reviewer, I cannot answer the question. It is an explorative study. Time will show us.

Comment: What were the main inclusion and exclusion criteria of this study? Please add an inclusion and exclusion criteria diagram.

Comment: Including inclusion and exclusion criteria in a diagram seems to be better. Please add a diagram about inclusion and exclusion criteria.

Response: Dear reviewer, I agree. I have added a diagram with the above mentioned criteria.

Comment: Due to Tables 1 and 2, is the role of gender important in this study?

Response: Dear reviewer, I have added the p- values in all the tables. According to our results gender is not a significantly parameter.

Comment: According to the Tables mentioned in this article, please check the style of the Tables of this article with the author's guideline of Biomedicines.

Response: Dear reviewer, you are right. I have rewritten the tables according to the author's guideline.

Comment: In order to improve the clarity of the introduction, it is recommended that you include some of the following sources as references.

Response: Dear reviewer, I have changed some and added new references from the list below:

  • https://doi.org/10.3390/biomedicines13020516
  • https://doi.org/10.3390/biomedicines13020360
  • https://doi.org/10.3390/biomedicines12122831
  • https://doi.org/10.3390/diagnostics15060781
  • https://doi.org/10.3390/pathogens14030262s

Round 2

Reviewer 1 Report

Comments and Suggestions for Authors

All of my comments have been addressed by authors. This manuscript can be accepted for publication. 

Reviewer 3 Report

Comments and Suggestions for Authors

Thanks